# Resumption of Cyclic Ovarian Activity by Herbal Preparation AyuFertin in Bulgarian Murrah Buffaloes at Early Postpartum

**DOI:** 10.3390/ani11020420

**Published:** 2021-02-06

**Authors:** Yordanka Ilieva, Nasko Vasilev, Ivan Fasulkov, Pencho Penchev, Desislava Abadjieva, Vanya Mladenova, Ayla Ilyazova, Dasha Mihaylova, Stanimir Yotov, Elena Kistanova

**Affiliations:** 1Agricultural Institute—Shumen, AA, 9700 Shumen, Bulgaria; jordani64@abv.bg (Y.I.); pen.penchev@gmail.com (P.P.); 2Department Obstetrics, Reproduction and Reproductive Disorders, Faculty of Veterinary Medicine, Trakia University, 6000 Stara Zagora, Bulgaria; nasvas@abv.bg (N.V.); i.fasulkov@gmail.com (I.F.); stanrad@abv.bg (S.Y.); 3Institute of Biology and Immunology of Reproduction, Bulgarian Academy of Sciences, 1113 Sofia, Bulgaria; dessi_l@abv.bg (D.A.); vanya_mladenova@abv.bg (V.M.); 4Department of Microbiology, University of Food Technologies, 4002 Plovdiv, Bulgaria; ayla_ilyazova@abv.bg; 5Department of Biotechnology, University of Food Technologies, 4002 Plovdiv, Bulgaria; dashamihaylova@yahoo.com

**Keywords:** postpartum buffalo, AyuFertin treatment, ovarian activity

## Abstract

**Simple Summary:**

Shortening the inter-calving interval is the most important problem in buffalo breeding, especially on organic farms, where the use of hormones is prohibited. The demand for herbal preparations that stimulate the activity of ovaries, as well as the choice of an appropriate time for their application, remains essential today. This study provides the results of the positive effect of the herbal product, which identified a wide range of bioactive substances—carotenoids, flavonoids, and fatty acids, including esters of trienoic fatty acids, on the resumption of the cyclic ovarian activity in early postpartum buffaloes. Overcoming postpartum anestrus in buffaloes is highly cost-effective due to cost savings for keeping dry, unproductive animals.

**Abstract:**

This study evaluates the phytochemical composition and ability of herbal preparation AyuFertin, supplemented shortly after calving, to resume the cyclic ovarian activity in postpartum Bulgarian Murrah buffaloes. In total, 13 healthy Bulgarian Murrah buffaloes at the age of 4–10 years, calved in March–April 2019 were involved in the experiment. Seven experimental buffaloes were treated individually per os with AyuFertin according to producer instruction. All animals were subjected to regular ultrasound examination of ovaries. GC-MS analysis of fatty acids composition and HPLC-DAD analysis of carotenoid and tocopherol content in AyuFertin were conducted. The determination of estrus occurrence and natural mating were done by a fertile bull. The development of the large follicles (12.5–14.3 mm) in 85.7% of treated animals was observed on day 24 postpartum compared to 0% in controls. Clinical signs of estrus were recorded in 71.4% of the animals, followed by an 80% pregnancy rate versus 16% in controls within 70 days after calving. In conclusion, the supplementation of the bioactive herbal product AyuFertin from day 21 postpartum can stimulate the follicular growth in the buffalo’s ovary, which leads to the development of follicles with ovulatory capacity, followed by successful ovulation. The above-mentioned treatment resulted in a better pregnancy rate in the treated multiparous compared to the control buffaloes bred by natural service.

## 1. Introduction

Due to low reproductive efficiency, overcoming the postpartum anestrus and shortening the inter-calving interval are the most important problems in buffalo breeding. These challenges cause heavy economic losses related to the cost of keeping dry, unproductive animals. The application of hormonal therapy for this purpose provides good results [1,2,3]. However, other than the high cost, the frequent use of exogenous hormones can impair the reproductive system of animals and lead to increased levels of sex hormones in milk, which affects consumer health [4,5,6]. In addition, in organic farms, the use of hormones is prohibited. Several herbal protocols instead of hormones or in combination with them have been developed for buffaloes to overcome the postpartum anestrus [7]. However, the main effort has been focused on the long-time postpartum anestrus—more than six months. Some authors report about the positive effect of Prajana HS, a commercial nonhormonal herbal preparation that affords optimization of ovarian functions and thus induces timely estrus, ovulation, and conception in treated buffaloes with true postpartum anestrus (6–8 months after calving) compared to the control, Prajana-free animals [8,9,10]. According to Chaudhiry et al. [11], the plants *Randia dumetorum* (fruit) and *Tinospora cordifolia* (stem) and their combination have an effective role in the treatment of anoestrus buffalo heifers. These plants induced estrus in more than 80% of the treated animals and were more effective than Prajana HS [11]. The mixture of *Abroma augusta* (root) and *Nigella sativa* (seed) in a 2:1 ratio also provoked a 50% estrus response in anoestrus buffaloes [12]. *Murraya koenigii* and *Aegle marmelos* have also shown a positive effect on overcoming the anestrus in buffaloes. After nine days of treatment, estrus was induced in 60% (6/10) of the animals followed by 66.67% (4/6) pregnancy [13].

The used herbs contain various biologically active substances with a positive effect on the reproductive system [10]. Among them, the role of the specific fatty acids involved in the biosynthesis of prostaglandins is important. Prostaglandins (PG), belonging to the eicosanoids group, are 20-carbon unsaturated hydroxyl fatty acids with a cyclopentane ring. The main precursor of PG, the arachidonic acid, is a product of the conversion of the linoleic essential fatty acid [14]. According to the literature, inadequate production of endogenous prostaglandin in the postpartum period leads to delay in uterine involution [15,16]. At the same time, the administration of prostaglandin F2-alfa in the early postpartum leads to accelerated uterine involution, to resumption of the fertile ovarian cyclicity, and to increased conception rate of first service [17,18,19].

The herbal preparation AyuFertin, suggested for heat induction in animals, includes the following herbs: *Piper longum, Piper nigrum, Zingiber officinale, Citrullus colocynthis*, which are well known in alternative medicine as substances with antioxidant, anti-inflammatory, and immunomodulatory properties [20]. Recent data show that these herbs contain various fatty acids [21,22].

Scientific data on the use of herbal preparations with natural prostaglandin precursors in the early stages of postpartum in buffaloes are very scarce, especially for AyuFertin herbal preparation, for which it is completely lacking. Our study aimed to investigate the phytochemical composition of AyuFertin and evaluate its effect when supplemented shortly after calving on the ovarian activity in the postpartum Bulgarian Murrah buffaloes.

## 2. Materials and Methods

### 2.1. Analysis of the AyuFertin Phytochemical Composition

#### 2.1.1. Carotenoid and Tocopherol Contents Evaluation

Qualitative and quantitative determinations of carotenoids and tocopherols were performed by using Elite LaChrome (Hitachi, Tokyo, Japan) high-performance liquid chromatography (HPLC) system equipped with diode array detector (DAD) and ELITE LaCHrome (Hitachi, Tokyo, Japan) software. The content of carotenoids was analyzed by the method of Georgieva et al. [23] with some modifications and the tocopherol content was assessed by the method described by Mihaylova et al. [24]. The results for both assays were expressed as μg/g dry weight (dw).

#### 2.1.2. Total Polyphenolic Content (TPC), Total Flavonoid Content (TFC), and Antioxidant Activity Evaluation

Representative samples of four grams of AyuFertin (randomly selected from four packages each of 500 g) were added to 20 mL of 80% methanol and sonicated at a frequency of 35 kHz with a maximum input power of 240 W, for 60 min, at 30 °C (UST 5.7150 Siel, Gabrovo, Bulgaria). The obtained extracts were filtered and stored at 4 °C without adding any preservatives until analyses.

##### Evaluation of Total Polyphenolic and Total Flavonoid Contents

The total polyphenolic content was analyzed following the method of Kujala et al. [25] with some modifications [26]. The TPC was expressed as mg gallic acid equivalents (GAE) per g dry weight (dw).

The total flavonoid content was evaluated according to the method described by Mihaylova et al. [26] and the results were expressed as mg quercetin equivalents (QE)/g dw.

##### Determination of Antioxidant Activity


*DPPH^•^ Radical Scavenging Assay*


The ability of the sample to donate an electron and scavenge 2,2-diphenil-1-picrylhydrazyl (DPPH) radical was determined by the slightly modified method of Brand-Williams et al. [27] as described by Mihaylova et al. [26]. The DPPH radical scavenging activity was presented as a function of the concentration of Trolox—Trolox equivalent antioxidant capacity (TEAC) and was defined as the concentration Trolox having equivalent antioxidant activity expressed as the μM per g dw (μM TE/g dw).


*ABTS^•+^ Radical Scavenging Assay*


The radical scavenging activity of the sample against 2,2’-azino-bis (3-ethylbenzothiazoline-6-sulfonic acid) (ABTS^•+^) was estimated according to Re et al. [28] and the results were expressed as TEAC value (μM TE/g dw).


*Ferric-Reducing Antioxidant Power (FRAP) Assay*


The FRAP assay was carried out according to the procedure of Benzie and Strain [29] and the results were expressed as μM TE/g dw.


*Cupric ion Reducing Antioxidant Capacity (CUPRAC) Assay*


The CUPRAC assay was carried out according to the procedure described by Apak et al. [30]. The results were expressed as μM TE/g dw.

#### 2.1.3. Gas-Chromatographic-Mass-Spectral (GC-MS) Analysis of Fatty Acids Composition

The extraction of the fatty acids was carried out by means of diethyl ether in a Soxhlet apparatus following the AOAC Method 920.39 [31]. The fatty acids methyl esters (FAMEs) were prepared in accordance with ISO 12966-2:2017 [32]. GC-MS analysis of the whole fatty acid content was performed using a Thermo Scientific GC-MS system (Thermo Scientific, Waltham, MA, USA) comprising an AI/AS 1310 auto-sampler and a Gas Chromatograph (Trace 1300) interfaced to an ISQ mass spectrometer, equipped with a TR-5MS fused silica capillary column (30 m × 0.25 mm, ID 0.25 µm). For GC-MS detection, electron impact mode was used. Mass spectra were taken at 70 eV, a scan interval of 0.2 s, and m/z range 46 to 650 Da. Helium gas was used as a carrier gas at a constant flow rate of 1 mL/min, and an injection volume of 1 μL was employed (a split ratio of 1:25). The MS transfer line temperature was set to 260 °C and the ion-source temperature was 220 °C. The oven temperature was programmed from 110 °C (held for 3 min), with an increase of 10 °C/min to 220 °C, held for 6 min, then 15 °C/min to 310 °C and held for 5 min. The injector temperature was maintained at 240 °C. The identification of the individual FAMEs was made by comparison of their mass spectra with the NIST standard reference database. Quantification of the identified fatty acids was made by the area normalization method (represented as % of the area of total fatty acids in the sample).

### 2.2. Animals Management and Experimental Design

The experiment was conducted in 13 clinically healthy postpartum Bulgarian Murrah buffaloes aged between 4 and 10 years with an average body weight of 580 ± 15.1 kg, calved in March–April and housed on the farm of Agricultural Institute—Shumen, Bulgaria. The animals were allotted two groups, control (*n* = 6) and experimental (*n* = 7) by the analog method, regarding body condition, and included both primiparous and multiparous buffaloes. The experiment was carried out during March–July, 2019.

The farming system was intensive—tie-stalls with exercise yard. All animals received the standard daily diet for dairy buffaloes. The ration components were as follows: wheat 15%, barley 12%, corn 56%, wheat bran 10%, sunflower oilcake 5%, dicalcium phosphate 0.6% salt 0.4%, and chalk 1%, providing 1629 kcal energy and 96 g digestible protein. Water intake was *ad libitum.*

The experiment was scheduled to start on day 21 after calving (Figure 1). The protocol included a three-day supplementation period and a secondary three-day supplementation if the individual buffalo did not express clinical signs of estrus and acceptance of bull within 10 days after the first treatment onset. The experimental animals were administered orally with feed supplement AyuFertin, containing powdered dried herbs (Indian Herbs Specialities Pvt. Ltd, Saharanpur, India), for three consecutive days in the dose of 3 g/100 kg body weight in accordance with the producer’s instructions from day 21 postpartum (d pp) in the first supplementation period and from 31d pp in the secondary one. From 21d pp, a fertile bull was introduced in both groups. Permanent observations for the presence of estrus and/or natural mating from bull were recorded until the end of the experiment.

The ultrasonography (USG) of the ovaries of all animals was performed on 20d pp, after the end of the first (24d pp) and second supplementation periods (34d pp) and 10 days after the end of the second supplementation period (44d pp). Ultrasound pregnancy check was carried out on days 70pp and 90pp. Transrectal ultrasonography was performed with a SonoScape A2 Vet (SonoScape Co. Ltd., Shenzhen, China), a multi-frequency (7–12 MHz) linear probe and transrectal approach. During the ovarian USG, the number and size of follicles and presence of a corpus luteum in one of the ovaries were registered.

The experimental work with animals on the farm was approved by the Bulgarian National Animal Ethics commission in accordance with the Bulgarian Veterinary Law (25/01/2011) regarding the living conditions and welfare of livestock animals used for experimental purposes, which is adapted to the European Union regulation 86/609 (AF 9747A-0002/N1430 from 05 April 2018).

### 2.3. Biochemical and Progesterone Analysis

The blood samples were collected from the jugular vein of all animals before (days 20 and 30) and after supplementations with AyuFertin (days 24 and 34) in vacuum blood collection tubes. After that, the samples were centrifuged, and the serum was separated in sterile Eppendorf tubes and stored at −20 °C until the time of analysis. The serum was analyzed for biochemical profiles including total protein (TP), blood glucose, total cholesterol (TChol), lactate dehydrogenase (LDH), and alkaline phosphatase enzymes (ALP). The evaluation of the mentioned parameters was performed using the standard kits for biochemical analyzer Mindray BA-88A (Mindray Bio-Medical Electronics Co, Ltd. Shenzhen, Nanshan, China).

The serum progesterone concentration was determined by the EIA method using the kit Progesterone EIA 96 TEST (Linear Chemicals, Spain). The intra-assay coefficient of variances ranged from 2.2 % to 7.1 % and the inter-assay coefficient of variance was 2.6% to 12.6 % for four repeated measurements.

### 2.4. Statistical Analysis

Due to the small number of used animals, the non-parametric statistic was applied. The results were analyzed using the software product Stat.Soft, v.10 (StatSoft Inc., Tulsa, OK, USA). The significance of the mean difference was defined by the non-parametric Mann–Whitney test. To compare the pregnancy rate between primiparous and multiparous animals, chi-square test for proportions with small samples was used [33]. The differences were considered significant at *p* < 0.05. The data are presented as a mean ± standard deviation (SD).

## 3. Results

### 3.1. Phytochemical Composition of AyuFertin

AyuFertin’s phytochemical analysis showed that it contained bioactive substances such as carotenoids, tocopherols, flavonoids, and fatty acids. Among the carotenoids and tocopherols found in the AyuFertin supplement, the distribution of δ-tocopherol, α-tocopherol, and lutein was determined (Table 1) with a predominance of α-tocopherol.

The established TPC and TFC in AyuFertin are a prerequisite for the antioxidant potential of the herbal preparation (Table 2).

The total lipid content was found to be 20.93% and the presence of trienoic fatty acid esters in addition to palmitic acid, linoleic acid, oleic acid, stearic acid, and palmitic monoglyceride was found (Table 3).

### 3.2. Effect of AyuFertin on the Blood Parameters

The main biochemical parameters of the blood did not differ significantly between the control and experimental groups at the beginning of the experiment (Table 4). AyuFertin treatment reduces the total serum cholesterol of the supplemented animals. The activity of the lactate dehydrogenase enzyme was enhanced, while the alkaline phosphatase did not change (Table 4).

The average progesterone level on day 21 did not differ significantly between the control and experimental groups (1.3 ± 1.55 ng/mL and 1.6 ± 1.75 ng/mL, respectively). The supplemented animals had a decreasing dynamic of progesterone after the treatments (Figure 2). The level of progesterone after the first supplementation decreased significantly (*p* < 0.01) compared to the pre-treatment period, while the decrease after the second AyuFertin treatment was not significant.

### 3.3. Effect of AyuFertin on the Ovarian Activity

The ovarian state of animals in both groups was similar on 20d pp (Table 5). In the control group, the number of small follicles did not change significantly with the advance of postpartum days, but the number of medium follicles enhanced more than twice. No large follicles were detected until day 24 pp (Appendix A). In this group, the presence of both, a large follicle and a corpus luteum, was found only on 44d pp in one animal. This animal came in heat and was served by the bull.

After the first treatment with AyuFertin the number of small and medium follicles decreased, while the number of large follicles increased (Appendix A). Two experimental animals exhibited estrus and were served by the bull. In five animals treated repeatedly, the number of small and medium follicles increased. Large follicles and corpora lutea were found in these animals in 80% and 60%, respectively (Table 5).

After the second treatment, three non-mated animals from the experimental group exhibited estrus and were served by the bull. In the other two animals, despite the determination of the large follicles with a size of 11–12.6 mm, no exhibition of estrus was detected. The resumption of cyclic ovarian activity in these animals was confirmed by the detection of corpora lutea during a later examination. In total, by ultrasound examination on the 70th day postpartum, three animals from the experimental group and one from the control group were diagnosed as pregnant (Appendix A).

During the control ultrasound examination, on the 90th day, one more pregnancy was found in the experimental group. Further analysis showed that all registered pregnancies in both groups (5/13) were in multiparous animals, while the pregnancies in primiparous and second-parity animals were not identified (Table 6).

Moreover, the positive effect of AyuFertin for MP buffaloes led to significant differences in pregnancy rate between MP and PP and SP buffaloes in the experimental group as well as between MP of experimental and MP of control animals (Table 6).

## 4. Discussion

The present study demonstrated a positive effect after early administration of the herbal supplement AyuFertin for overcoming the postpartum anestrus and shortening the intercalving period in Bulgarian Murrah buffaloes. The treatment with AyuFertin not only stimulated follicle growth and led to successful ovulation, but also increased the pregnancy rate in the treated group of animals by 40.5% compared to the controls within 70 days after calving. Four buffalo-calves were born within one year of calving (calving interval 362.7 ± 6.21 days) in the experimental group compared to only one in the control group.

No adverse effect of AyuFertin on the basic biochemical parameters in the blood serum of the supplemented animals was found. All post-treatment values, including the deviations in cholesterol and LDH, were in the physiological range of these parameters for buffaloes [34,35].

The main active ingredients of AyuFertin are believed to be trienoic acids. However, our biochemical analysis of AyuFertin showed that linoleic is the main fatty acid to be present in the sample (42% versus 14.3% for trienoic acids). Linoleic acid from the group of omega-6 is considered an essential fatty acid because animal and human bodies are unable to synthesize these compounds [36]. Several studies have reported that this essential fatty acid (EFA) serves as an in vivo precursor to arachidonic acid, whose main function is the production of prostaglandins of 2-series as PGF2α and E2 with importance in mammalian reproduction [37,38,39]. According to Arosh et al. [40], prostaglandins biosynthesis is selectively directed toward PGF2α during luteolysis and to PGE2 at the time of the establishment of pregnancy. Decreased progesterone levels in our experimental animals after treatment with AyuFertin, are likely to indirectly confirm increased PGF2α, which plays important role in stopping progesterone production, leading to the onset of a new estrous cycle and follicular growth [41]. Our data are supported by the results of Dirandeh et al. [39], who reported that feeding n-6 polyunsaturated fatty acids (PUFA) increases the endometrial percentages of linoleic and arachidonic acids (AA), enhances the synthesis of prostaglandin F2α (PGF2α), and improves the uterine health. The advantages of the natural precursors of PGF2α over the administration of a single high dose of exogenous PGF are systemic release and a lower dose of produced prostaglandin F, which more accurately reflect the physiologic amounts available during regular CL regression [42].

The fatty acid trienoic esters and alpha-linolenic acid (ALA) belonging to the n-3 PUFAs group were also found in AyuFertin. Animals and humans cannot synthesize trienoic acid on their own, such as linolenic acid (9,12,15-octadecatrienoic acid), therefore it is an essential part of the diet that should be obtained from plant sources in an appropriate quantity. After ingestion, this acid is metabolized to the important eicosapentaenoic acid (EPA C20: 5) and C22 (docosahexaenoic acid, 22: 6) acids, and C20 is a precursor to 3-series prostaglandins biosynthesis [37,38]. Decreased expression of the enzyme prostaglandin F synthase (PGFS), which converts prostaglandin H2 to PGF2a, has been reported in cows fed an n-3 PUFA diet, while expression of the enzyme prostaglandin E synthase (PGES), which in turn increases PGE2 synthesis, was enhanced [43,44]. Prostaglandin E2 plays a very important role during early pregnancy as a luteoprotective factor. It helps maintain CL and completely blocks the action of PGF at the CL level [41]. In addition to increasing the prostaglandin E2 synthesis, eicosapentaenoic acid (EPA,20:5) generates the less-inflammatory and less potent prostaglandin E3 (PGE3) [45], which could reduce the immunological problems of maternal pregnancy recognition. These data may explain our results, showing a high pregnancy rate (80%) among estrus-manifesting animals in the experimental group and a 100% delivery rate.

Therefore, not just are trienoic fatty acids reported as the main active substances in many herbal heat inducers [9,10,11], but the n-6 FA precursors of prostaglandins are also important for the successful application of the herbal preparation for overcoming anestrus. In fact, the presence of both n-3 and n-6 PUFAs in the AyuFertin preparation in an appropriate ratio stimulated the follicular growth to the dominant follicle and preserved the pregnancy.

It should be noted that the supplementation of AyuFertin is more beneficial for the multiparous buffaloes. All animals that became pregnant have had 4–6 parities. There is evidence [46] that parity affects the postpartum to estrus interval, which is longer in PP than in MP buffaloes, but the specific effect of AyuFertin on MP buffaloes is not yet clear and further studies in representative groups of animals with different parities are needed.

A detailed biochemical analysis of AyuFertin showed that in addition to specific fatty acids, the preparation contains many other active substances with an important role in reproduction. Among the tocopherols, high levels of alfa-tocopherol, known as vitamin E, have been found in the AyuFertin. Vitamin E is an important component of membranes in all cells, including gametes, and functions as an antioxidant to protect cellular membranes, for example by protecting polyunsaturated fatty acids (PUFAs) from autoxidation [47,48]. It also plays important role in the ovary protection against oxygen radicals during corpus luteum regression [49] and can affect fertility by improving ovulation rate and endometrial response [50,51]. The positive effect of vitamin E has been shown for buffalo oocytes in vitro. The supplementation of vitamin E to the culture medium increases the developmental competence of buffalo oocytes [52].

The noticeable increase of follicular growth and size established in 85.7% of the treated animals on 24d pp may also be provoked by other components of AyuFertin such as polyphenols and lutein. Some authors have reported that these substances may affect the expression of follicular growth factors as GDF9 and BMP15 leading to granulosa cell proliferation [53,54]. However, the functional maturity of granulosa cells sometimes delays, despite their intensive growth. The above-mentioned may explain our finding that not all experimental animals possessing large follicles exhibited estrus. The granulosa cells must produce a sufficient quantity of estrogens to show estrus [55]. As substances with high antioxidant and anti-inflammatory properties, polyphenols and lutein are important for maintaining the redox balance in the ovary, which is necessary for the proper oocyte development and maturation [56,57].

The aforementioned beneficial effects of alfa-tocopherol, polyphenols, and carotenoids are likely to provide additional value for the successful resumption of ovarian activity, ovulation, and fertilization of the experimental animals.

## 5. Conclusions

The supplementation of the bioactive herbal product AyuFertin from day 21 postpartum can stimulate follicular growth in the buffalo’s ovary, which leads to the development of follicles with ovulatory capacity, followed by successful ovulation. With the introduction of a fertile bull in both groups, the aforementioned treatment resulted in a better pregnancy rate in the MP buffaloes from the experimental group compared to the control group. However, the mechanism of the beneficial effect of AyuFertin in multiparous buffaloes remains to be elucidated in further investigations in a larger number of animals.

## Figures and Tables

**Figure 1 animals-11-00420-f001:**
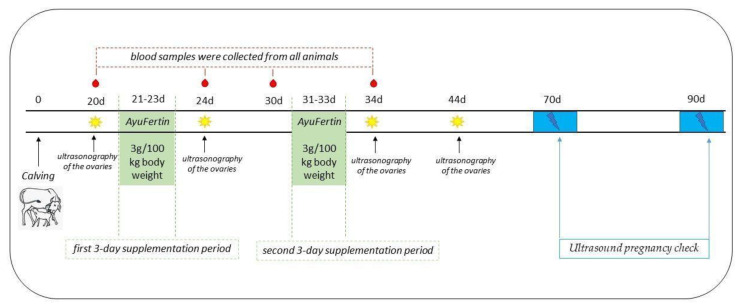
Scheme of the experimental design.

**Figure 2 animals-11-00420-f002:**
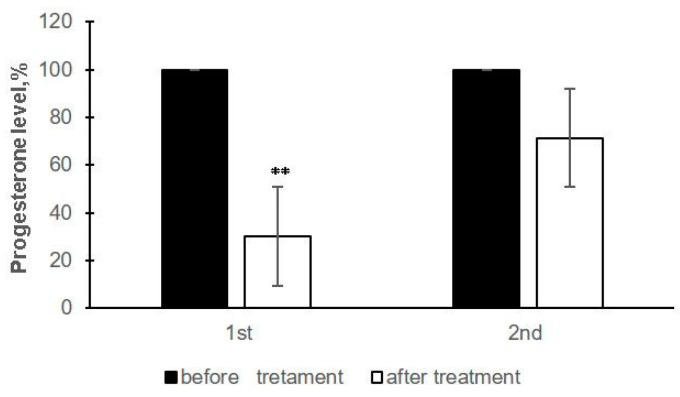
Relative change of progesterone level in blood serum of experimental Bulgarian Murrah buffaloes according to the number of treatments with AyuFertin supplement. The progesterone level before the first and second treatments was accepted as 100%. ** *p* < 0.01 compared to pre-treatment period.

**Table 1 animals-11-00420-t001:** Tocopherols (µg/g dw) and carotenoids (µg/g dw) content of AyuFertin.

Sample/Assay	AyuFertin
Tocopherols, µg/g dw	
δ-tocopherol,	6.37 ± 0.51
α-tocopherol	22.87 ± 1.11
ϒ-tocopherol	n.d *
Carotenoids, µg/g dw	
lutein	5.51 ± 0.45
lycopene	n.d
β-carotene	n.d

* n.d—not detected.

**Table 2 animals-11-00420-t002:** Total polyphenolic content (TPC, mg GAE/g dw), total flavonoid content (TFC, mg QE/g dw), and in vitro antioxidant activity (μM TE/g dw) of AyuFertin.

Sample/Assay	AyuFertin
Total polyphenolic content, mg GAE/g dw	21.00 ± 0.93
Total flavonoid content, mg QE/g dw	319.62 ± 9.83
In vitro antioxidant activity, μM TE/g dw	
ABTS	191.66 ± 2.52
DPPH	209.25 ± 1.71
FRAP	334.28 ± 0.70
CUPRAC	331.27 ± 5.94

**Table 3 animals-11-00420-t003:** Total lipid content and fatty acids composition of AyuFertin.

Apex RT	Area	% Area	Fatty Acids
14.76	77,575,835	8.81	Palmitic acid
17.23	371,305,357	42.15	Linoleic acid
17.32	253,137,987	28.73	Oleic acid
17.77	46,436,988	5.27	Stearic acid
20.38	67,170,678	7.62	CH_3_ esters with Mw 292: 6-cis,9-cis,11-trans-octadecatrienoate 9.cis.,11.trans.t,13.trans.-octadecatrienoate 9,12,15-Octadecatrienoic acidGamma-Linolenic acid
20.54	22,680,426	2.57	
20.84	15,537,705	1.76	
21.29	21,773,425	2.47	
24.2	3,012,782	0.34	Palmitin monoglyceride
Total lipid content		20.93	

**Table 4 animals-11-00420-t004:** Biochemical parameters in the blood serum of control and treated with AyuFertin Bulgarian Murrah buffaloes.

Group	Biochemical Parameters				
	TP g/dL	Glucose mg/dL	TChol mg/dL	LDH U/L	ALP U/L
Control (*n* = 6)					
	6.5 ± 0.78 ^a^	38.7 ± 3.2 ^a^	161.5 ± 5.03 ^a^	527.3 ± 23.26 ^a^	188.5 ± 14.54 ^a^
Experimental (*n* = 7)					
BFT (*n* = 7)	5.9 ± 0.47 ^b^	36.1 ± 2.32 ^b^	157 ± 8.45 ^b^	550 ± 34.4 ^b^	178 ± 12.05 ^b^
AFT (*n* = 7)	5.4 ± 0.49 ^c^	38.4 ± 2.51 ^c^	142.7 ± 4.42 ^bc^	560 ± 47.1 ^c^	177.3 ± 16.47 ^c^
BST (*n* = 5)	5.4 ± 0.28 ^d^	38.3 ± 2.48 ^d^	147 ± 4.78 ^bd^	592 ± 16.75 ^bd^	169.4 ± 17.7 ^d^
AST (*n* = 5)	6.8 ± 0.87 ^e^	36.9 ± 2.20 ^e^	153.4 ± 10.89 ^e^	628.3 ± 44.51 ^be^	177.8 ± 16.16 ^e^

^a,b,c,d,e^ Superscripts mark the means of different groups. ^bc,bd,be^ Means with two different superscripts within the same column differ at *p* < 0.05; BFT—before first treatment; AFT—after first treatment; BST—before second treatment; AST—after second treatment; TP-total protein; TChol—total cholesterol; LDH—lactate dehydrogenase; ALP—alkaline phosphatase.

**Table 5 animals-11-00420-t005:** Ovarian structures, estrus expression, and pregnancy rate in control and experimental Bulgarian Murrah buffaloes.

Parameters	Control Group (*n* = 6)				Experimental Group (*n* = 7)			
Postpartum days	20	24	34	44	20	24	34	44
Ovarian structures								
Small follicles (<6 mm) % (*n*)	100 (6/6)	100 (6/6)	100 (6/6)	100 (6/6)	100 (7/7)	42.8 (3/7)	60 (3/5)	100 (7/7)
Medium follicles (6–9 mm) % (*n*)	33.4 (2/6)	33.4 (2/6)	83.3 (5/6)	100 (6/6)	42.8 (3/7)	42.8 (3/7)	100 (5/5)	100 (7/7)
Large follicles (≥10 mm) % (*n*)	0 (0/6)	0 (0/6)	16.6 (1/6)	33.4 (2/6)	0 (0/7)	85.7 (6/7)	80 (4/5)	71.4 (5/7)
Corpus luteum % (*n*)	0 (0/6)	0 (0/6)	0 (0/6)	16.6 (1/6)	0 (0/7)	14.2 (1/7)	60 (3/5)	42.8 (3/7)
Recorded estrus with bull mating % (*n*)								
After 1st treatment					28.6 (2/7)			
After 2nd treatment	16.6 (1/6)				60.0 (3/5)			
Total	16.6 (1/6)				71.4 (5/7)			
Pregnancy rate, % (n)								
Day 70	16.6 (1/6)				42.9 (3/7)			
Day 90	16.6 (1/6)				57.1 (4/7)			

**Table 6 animals-11-00420-t006:** Comparative analysis of the biometric parameters and established pregnancies between the control and experimental groups.

Parameters	Control Group (*n* = 6)			Experimental Group (*n* = 7)			*p*-Value between Groups
Calving period	14 March–22 April 2019			29 March–29 April 2019			
Average age (years)	7 ± 2.4			6.6 ± 2.3			NS
Average body weight (kg)	583.3 ± 18.9			577.1 ± 15.1			NS
Average number of lactations	3.8 ± 1.79			3.4 ± 2.07			NS
Parity	PP	SP	MP	PP	SP	MP	
	1/6	1/6	4/6	2/7	1/7	4/7	NS for all
Pregnancies	0/1	0/1	1/4	0/2	0/1	4/4	0.04 for MP
	PP vs. MP	SP vs. MP	PP + SP vs.MP	PP vs. MP	SP vs. MP	PP + SP vs. MP	
	*p* = 0.61	*p* = 0.61	*p* = 0.47	*p* = 0.02	*p* = 0.04	*p* = 0.01	

PP—primiparous animals; SP—second parity animals; MP—multiparous animals; NS—non-significant.

## Data Availability

All data are included in the present article and its Appendix A.

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
