# Peer review of "Resumption of Cyclic Ovarian Activity by Herbal Preparation AyuFertin in Bulgarian Murrah Buffaloes at Early Postpartum"

_animals, 2021, doi:10.3390/ani11020420_

Round 1
Reviewer 1 Report
The manuscript reports data on the phytochemical composition of a commercial polyherbal formulation and its ability to resume the cyclic ovarian activity in Bulgarian Murrah buffaloes after calving.
The topic is certainly interesting, as the inter-calving interval is one of the most important problems in buffalo breeding. In addition, the identification of natural substances, such as polyherbal formulations of sure efficacy, can represent a valid tool to be used also but not only in organic farming.
In my opinion, the paper should be accepted for publication only after major revision, possibly as short communication, as the small number of subjects used for the experimentation and some limitations in the analysis of the polyherbal compound suggest that further investigations with a large number of animals are necessary to clarify the beneficial effect of this polyherbal formulation. In addition, the standard diet was not characterized in its nutraceutical components, as done for the polyherbal formulation.
M&M
The polyherbal preparation should be better described. No indication concerning the producer or the actual composition (% of each herb, powdered dried herb or extracts, etc.) is reported.
Regarding the extraction method, several solvents are used according to the different substances to be extracted. Since no reference is given, the authors must justify why they used methanol (80%) as a solvent followed by sonication. In addition, the authors must clarify whether the extracts were dried or left in solvent until analysis.
The phytochemical characteristics are expressed as g of dry weight. Does this refer to the dry extract or to the polyherbal product?
From what reported, it seems that the authors have analysed only one package of polyherbal product. In this case, how do the authors come up with a mean and a standard deviation?
Results
All the figures regarding he USGs of ovary and pregnancy can be presented as supplementary materials.
Figure 1. I probably did not understand, but progesterone is indicated as a percentage and not as a concentration; moreover, the SD is not reported. Can the authors clarify this better?
Discussion
I am aware of the difficulties in discussing results obtained with a natural product containing multiple substances with nutraceutical activity. Nonetheless, it allows me to highlight how the discussion is mainly based on linoleic, the main fatty acid present in the polyherbal formulation, and on trienoic acids. Authors reported several information from the literature, often not discussing their results or sometimes just reporting hypotheses or speculating on them. In my opinion the discussion should be rewritten considering these criticisms.
Author Response
Please, see the attachment

Reviewer 2 Report
Review
Animals - 1062870
Resumption of cyclic ovarian activity by the natural precursors of prostaglandins from AyuFertin in Bulgarian Murrah buffaloes at early postpartum
General comments:
The manuscript deals with the influence of the herbal preparation “AyuFertin” on the resumption of cyclic activity in postpartum Bulgarian Murrah buffaloes. It is an interesting and new contribution that falls within the scope of the journal. However, there is one major concern that has to be addressed before this manuscript can be published.
From this reviewer’s point of view, the low number of animals used in this study is not necessarily suitable to detect differences in the resumption of ovarian cyclicity and pregnancy rates between buffaloes with and without feeding of AyuFertin. The crucial point is the homogeneity of both groups regarding age, body weight and parity of the animals. The influence of these confounding factors should be statistically analyzed. At least, the average age, body weight and number of calvings should be provided for each group (experimental and control group) and a comparison between primi- and multiparous buffaloes regarding the reproductive results should be added.
Specific comments:
- Line 24: “resumption”
- Line 25: “in early postpartum buffaloes”
- Line 29: Add a space between the two sentences. Please check for missing spaces also the whole manuscript, especially lines 111, 189, 190, and 202.
- Line 33: Use “AyuFertin” consistently throughout the manuscript.
- Lines 37-40: To draw these conclusions, you have to provide results about large follicles and ovulations in control buffaloes before.
- Line 38: “from 21 day postpartum”
- Lines 40-41: A fertile bull was also introduced in the control group. Therefore, it is the treatment that results in a better pregnancy rate in treated compared to control buffaloes after the introduction of a fertile bull in both groups. Please rephrase.
- Line 47: “challenges” instead of “problems”
- Lines 50-51: Please specify how hormones effect the milk quality (the reader does not even get any information if effects are positive or negative) and what other side effects are meant.
- Line 54: “the main effort has been focused”
- Line 57: What is “true postpartum”?
- Line 58: Please define “control animals” (Same time after calving but not fed with Prajana HS?)
- Line 71: double space between “is” and “a”? Please check the whole manuscript for double spaces, especially lines 152, 158, 166, 200, 201, 202, 294, 306, 323, and 324.
- Line 80: “recent”
- Line 83: “AyuFertin”
- Line 98: “[22]” instead of “(22 2018)”
- Line 109: “dw” is already defined in line 99.
-Line 149: Please add a graphic time schedule for the experimental design.
- Line 153: “allotted”
- Line 154: Please provide the number of primiparous and multiparous buffaloes in each group, as well as the average body weight and average age (including standard deviation) for each group.
- Lines 157-159: What does the “-“ before the percentage stand for? It is used before all percentages except dicalcium phosphate?!
- Line 160: “ad libitum”
- Line 167-168: “from postpartum day 21 on” and “from postpartum day 31 on” instead of “since”.
- Line 168: “At the start of treatment…” Additionally, it is unclear if Day 21 and/or Day31 is meant – please clarify.
- Line 168-169: “start of treatment” and “in both groups” is mutually exclusive, since the control group is not treated! Please indicate the days, when the bull was introduced in the treated and control group, respectively.
- Line 171-172: The abbreviation “postpartum day (PPD)” should be defined when “postpartum day” was used for the first time in this manuscript. If this abbreviation is defined, only the abbreviation should be used afterwards.
- Line 198: A statistical analysis of all reproductive results comparing primiparous and multiparous buffaloes is mandatory!
-Line 210: Delete “22.87 ± 1.11 µg/g dw”. To mention this result in the sentence is redundant because it is written in Table 1.
- Lines 213-214: Delete the information in brackets because these results are mentioned in Table 2.
- Table 2: I suggest to broaden the first column of this table because the line break within the units is unfavorable.
- Table 4: Delete the comma after “Glucose”. Please use periods instead of commas for all numbers consistently! Furthermore, it is hard to believe that most results within the columns differ significantly, particularly if you take the small number of animals into consideration. Please check again statistically, especially for TP in AFT and BST, for glucose in AFT and BST, and for ALP in AFT and AST.
- Line 235: I would suggest not to write “similar dynamic”, since the grade of decrease in progesterone is very different after first and second treatment.
- Lines 237-238: The sentence “This tendency…” does not make sense. Please rephrase.
- Figure 1: Please add error bars!
- Line 244: “at day 20 postpartum”
- Line 247: “(Figure 2)” should be deleted and written after the next sentence: “…until day 24 (Figure 2).”
- Lines 247-248: “…the presence of both, a large follicle and a corpus luteum, was found…”
- Table 5: There is no information about bull mating in the control group at the time of treatments in the experimental group. Please add if the information is available. Please also add the information that you provide the pregnancy rate on day 90 postpartum.
- Lines 252-253: Please delete the sentence because it is aforementioned.
- Line 254: Only one animal came in heat. So, the other one with a large follicle was not in heat?
- Line 259: Two animals exhibited estrus, although six had large follicles? Isn’t this unexpected? Please discuss.
- Line 262: Delete the period before the bracket.
- Line 275-276: Was the one pregnancy more due to later mating of this animal or due to a false negative pregnancy diagnosis at 70 days postpartum?
- Line 277: The phrase “in primiparous animals, between 4 and 6 lactations” is a contradiction!
- Line 285: “approximately 41%” or “40.5%”
- Line 300: “Arosh et al. [37]”
- Line 306: “Dirandeh et al. [36]”
- Line 311: “available”
- Line 314: “… AyuFertin. Animals and humans cannot produce …”
- Line 333: The 80% (4/5) are the animals with large follicles at 34 days postpartum. It is not clear from Table 5 that they were all pregnant. Information regarding pregnancies should be extended.
- Line 336: “All animals that became pregnant have had 4 to 6 parities before."
- Line 347-348: “buffalo oocytes”
- Lines 359-361: The introduction of a fertile bull was also in the control group. Therefore, the treatment seems to be responsible for better pregnancy rates. Please rephrase, for example: “Introducing of a fertile bull in both groups, the above-mentioned treatment results in a better pregnancy rate in the experimental compared to the control group.
- Line 362: It is incomprehensible for the reader why you mention multiparous buffaloes specifically in the last sentence. It should become clear from the conclusions that the reproductive improvements confine to multiparous animals.
Author Response
Please, see the attachment

Round 2
Reviewer 1 Report
The quality of the manuscript has been improved and it is acceptable for publication in the present form.
Author Response
Dear Reviewer 1,
We highly appreciate your positive evaluation of our response and importance of work, which we have done for the manuscript improving. Thank you for the decision that quality of our manuscript has been improved and it is acceptable for publication in the present form.
Sincerely yours,
Dr. E. Kistanova